# How does digital technology affect manufacturing upgrading? Theory and evidence from China

Qingwei Fu[1,2]*

1 School of International Economics and Trade, Shandong University of Finance and Economics, Jinan, China, 2 School of Economics and Trade, Shandong Management University, Jinan, China

* qingweifu@126.com

## Abstract

Digital technology becomes the new engine of manufacturing upgrading. The article brings digital technology and manufacturing in global value chain into the same analytical framework, measures development level of digital technology by using World Input-Output Database data, Theoretical analysis shows that impact of digital technology on manufacturing upgrading has innovation effect, resource allocation effect and penetration effect. Empirical test results show that (1) Digital technology level index of China's manufacturing industry increased from 0.286 in 2001 to 0.359 in 2014, The effect of digital technology on upgrading of Chinese manufacturing industry is significant positive at 5% level with influence coefficient of 0.129; Distinguish digital technology sources found that domestic digital technology is used in a large proportion, foreign digital technology is more efficient, the influence coefficients are 0.124 and 0.703 respectively, and both promote upgrading of manufacturing industry; (2) The role of digital technology varies among industries. The promotion effect on capital intensive industries and technology intensive industries is positive at 5% level, and influence coefficients are 0.124 and 0.108 respectively. It has significant positive impact on low-to-medium and medium-to-high technology manufacturing at 1% level with influence coefficients of 0.177 and 0.138 respectively. Therefore, China needs to accelerate deep integration of digital technology and manufacturing industry, and climb to the middle and high end of global value chain.

**Citation:** Fu Q (2022) How does digital technology affect manufacturing upgrading? Theory and evidence from China. PLoS ONE 17(5): e0267299. https://doi.org/10.1371/journal.pone.0267299

**Data Availability Statement:** All relevant data are within the paper and its Supporting Information files.

**Funding:** The author(s) received no specific funding for this work.

## 1. Introduction

Since the 21st century, digital technologies represented by artificial intelligence, block chain, cloud computing, big data have flourished, accelerated penetration into various fields, promoted deep integration and innovation of industries, and globalization has entered the era of digital economy. The digital wave has set off a global scientific and technological revolution, It is quietly promoting the change of industrial structure. Digital technologies will profoundly change competitive advantages of various countries and reconstruct the world economic pattern [1]. The combination, iteration, integration and innovation of digital technology and

**Competing interests:** The authors have declared that no competing interests exist.

traditional production factors not only triggered optimization and transformation of production factors, systematic and revolutionary group breakthroughs, but also gave birth to new technology, new capital and new labor such as artificial intelligence, financial technology and intelligent robots, driving transformation of traditional industries to networking and intelligence [2]. Data has become a new key factor of production in manufacturing industry, a new engine to release potential of economic growth, accelerate diffusion of knowledge, optimize allocation of resources [3]. In 2019, the added value of China's digital economy reached 35.8 trillion-yuan, accounting for 36.2% of GDP. According to comparable standards, the nominal growth of digital economy was 15.6%, about 7.85% higher than the nominal growth of GDP in the same period. The position of digital economy was further highlighted, digital added value of China's industry was about 28.8 trillion-yuan, accounting for 29% of the GDP, and industrial economic penetration rate was 19.5%. The proportion of software and information technology service industry, Internet and service industry in the industry continues to increase, which promotes continuous optimization of internal structure of digital industry, and proportion of telecommunications industry and electronic information manufacturing industry continues to decline [4]. Digital development of China's manufacturing industry is shown in the Table 1 below.

The progress of digital technology showed an exponential growth trend. It has characteristic of reshaping production mode, organization mode, service mode and innovation mode. It has become strategic commanding point of industrial development in various countries, and has been highly valued by governments of various countries [5], such as "Industrial Internet" of the United States and "Industry 4.0" in Germany are all trying to promote deep integration of information technology and industry.

After 40 years of rapid development, China's manufacturing industry has established an industrial system with complete categories and the scale ranks first in the world, but it is far from a strong manufacturing power. It has been at the low end of the global value chain (GVC) for a long time. It is deeply troubled by the need to import core technologies and key parts from abroad [6]. With increase of resource constraints and weakening of labor cost advantages, The model of participating in global division of labor with primary elements is no longer sustainable. Facing dual challenges of "high-end manufacturing return" in developed countries and "low-end industrial competition" in developing countries, it has become an urgent issue to jump out of the trap of functional division of labor [7]. Empowering with digital technology, extending to innovative Research and Development(R&D) and service terminals with high added value, and accelerating formation of new competitive advantages have become a major issue that cannot be delayed [8]. After the "COVID-19 epidemic impact", China's digital economy has accelerated, shifted gears and developed at full speed, Digital technology will promote efficient and accurate connection between supply and demand, and promote "separate aggregation" of producer services. China is establishing a "dual circulation" development pattern in which domestic economic cycle plays a leading role while international

**Table 1. Digital development of China's manufacturing industry.**

| Year | 2016 | 2017 | 2018 | 2019 |
|---|---|---|---|---|
| Scale of digital economy (trillion-yuan) | 22.6 | 27.2 | 31.3 | 35.8 |
| Proportion in GDP | 30% | 32.9% | 34.8% | 36.2% |
| Proportion of industrial digitalization in GDP | 21% | 24.9% | 27.6% | 29% |
| Industrial economic penetration | 17% | 17.2% | 18.3% | 19.5% |

Data source: White paper on the development of China's digital economy (2020), CAICT Publications, 2020.

economic cycle remains its extension and supplement [9, 10]. Digital innovation opens a new world for development of manufacturing industry. The infiltration of digital technology and manufacturing industry has not only brought about improvement of efficiency, but also spawned new products, new formats and new models. The technological transformation and upgrading of the industry promote global value chain to expand digitally [11]. The parallel advancement and integration of manufacturing industry along technological route of digitization, networking and intelligence is the main way to realize intelligent manufacturing, and it is also main direction of the manufacturing power [12, 13]. Digital transformation of manufacturing is the key to achieve industrial transformation. So, what is the current development level of digital technology in China's manufacturing industry? What is the impact of digital technology on upgrading of Chinese manufacturing in global value chain?

This article will try to use data of World Input-Output Database (hereinafter referred to as WIOD) to calculate development level of digital technology in China, Then theoretical analysis and empirical test its impact on manufacturing upgrading, and on this basis, put forward development strategies and policy suggestions. Rest of the paper is organized as follows: Section 2 is literature review, which sorts out the latest research of domestic and foreign scholars in the field; Section 3 theoretical analyses impact mechanism of digital technology on manufacturing upgrading; Section 4 constructs an empirical model to test impact of digital technology on manufacturing upgrading; Section 5 is test results and analysis; Section 6 concludes.

## 2. Literature review

With the rise of intelligent robots and "Internet +", digital revolution has provided a valuable opportunity for China's manufacturing industry to leap from "following" to "running side by side" and then to "leading". Chinese and foreign literature have studied concept and function of digital technology from different angles, Due to rapid development of digital technology, its concept has not yet formed a unified standard worldwide. The U.S. Bureau of Economic Analysis (BEA) has defined scope of digital economy, and calculated scale of added value and total output of the U.S. digital economy [14]; Based on the OECD conceptual framework of digital economy, Statistics New Zealand estimates that total output of digital ordered products accounted for 20% of total output of the national economy from 2007 to 2015 [15]. In the "White Paper on China's Digital Economy Development (2020)", digital technology includes telecommunications industry, electronic information manufacturing, software and information technology service industries, the Internet and related service industries. According to the classification of OECD input-output table and definition of Economic and Statistics Administration (ESA) of the U.S. Department of Commerce, digital technology was summarized into information and communication technology industry and mechanical automation industry [16]. The second is to discuss impact of digital technology on manufacturing upgrading from a theoretical perspective. Qualitative research generally affirms promoting role of digital technology. For example, The widespread use of digital technology produced economies of scale, economies of scope, long-tail effects, which could provide better matching mechanisms and innovation incentives [17]; It can expand division boundary of industrial chain, reduce transaction costs, enhance value-added space and force demand changes, thus emerging new manufacturing modes such as service-oriented manufacturing and networked collaborative manufacturing [18]; Digital technology was a powerful weapon to enhance dynamic capabilities of enterprises. Through digital transformation, the seamless connection between engineering chain and supply chain can be realized. The new production model can enhance international competitiveness of manufacturing industry [19]. The third is to test impact of

digital technology on upgrading of manufacturing industry from an empirical point of view. Due to limitations of digital technology calculation methods and data acquisition, more rigorous quantitative research is still lacking. Most studies are carried out at industry or national level based on input-output table. For example, OECD data was used to measure the level of digital technology, and verified the significant role of digital technology in promoting China's position in GVC [20]; WIOD data was used to analyze trade added value of various industries in manufacturing industry, and empirically tested reconstruction effect of digital technology on manufacturing industry [21]; There are also researches using province data and entropy method to calculate level of digital technology, and found that intelligence and digitization can promote and enhance upgrading of manufacturing industry [22, 23]. Research on micro level is carried out from perspective of specific digital technologies such as industrial robots and the Internet. For example, artificial intelligence, as a general digital technology, has attracted attention of scholars at home and abroad. Acemoglu et al. (2020) found that the use of industrial robots could supplement and replace labor, increased added value of enterprises, and promoted productivity and economic growth [24]. According to robot data provided by the International Robot Federation (IRF), it was found that artificial intelligence can improve participation and division of labor through technological innovation and optimization of resource allocation [25, 26]; The degree of enterprise Internet was measured based on data that enterprises have microblogs, mailboxes and home pages, and verified role of Internet in improving domestic added value rate of Chinese enterprises' exports [27]. Research at micro level deeply reveals development trend of digitization and networking of manufacturing industry. On the whole, empirical research describes digital technology from different perspectives, and it is difficult to show full picture of digital technology and integration of various types of digital technology with manufacturing.

The existing literature provides multi-dimensional perspectives and rich connotations. By reviewing the above literature, we can find that to explore mechanism of digital technology from perspective of global value chain, more detailed and rich research is urgently needed; Measurement of the level of digital technology is still in exploratory stage, Moreover, development level of China's manufacturing industry varies significantly between regions, the eastern region is significantly higher than other regions, followed by the central and western regions, and the northeast is the weakest, and majority of small and medium-sized enterprises are still in initial stage of digitalization [28]. Single indicator in most empirical studies ignores related close factors such as foundation of technology application [29]; Under GVC division of labor model, conclusion may be biased without distinguishing differences of digital input sources. In view of this, this article uses WIOD data to incorporate digital technology and manufacturing GVC into the same analysis framework, effectively distinguishing and empirically testing impact of domestic and foreign digital technologies on upgrading of China's manufacturing industry.

Compared with current research, marginal contributions of this article are as follows: **First,** Concerning the content, using the latest data in WIOD, the article calculates digital technology level of 18 sub-industries of China's manufacturing industry, It is comprehensive to assess impact of digital technology on upgrading of China's manufacturing industry; The national attribute of digital technology is fully considered, and domestic and foreign parts are distinguished according to the source of digital technology; Improved GVC position index is used to accurately measure status of China's manufacturing industry participating in GVC from an economic perspective, which makes up for defects of original GVC index and completely describes China's role in global value chain. **Second,** selection of research perspective is unique, the article analyzes impact of digital technology on China's manufacturing industry from perspective of global value chain, which enriches relevant research on digital economy and status of China's manufacturing industry; **Third** is about research methodology, starting

from heterogeneity of manufacturing factor intensity and technology level classification, the article distinguish heterogeneous impact of digital technology on manufacturing upgrading, deepens understanding of relationship between digital technology and manufacturing upgrading, and provides reliable theoretical and empirical evidence for China's manufacturing industry to take the road of digital transformation and upgrading.

## 3. Theoretical mechanism analysis

### 3.1 The innovation effect

Innovation is the biggest challenge facing upgrading of China's manufacturing industry [30]. Digital technology has changed innovation model of manufacturing enterprises. The original innovation process is long and irreversible from design, R&D to production and marketing, If application problems are encountered and then modified and perfected, the time and cost were extreme high. Digital technology breaks constraints of time and space [31]. For example, networked open innovation model can quickly conduct synchronous testing and staggered development of creative ideas, virtual technology and 3D printing technology can turn ideas into real objects, mobile factories become reality, and digital interconnection technology and global value chain embedding have a revolutionary impact on manufacturing innovation. Through the Internet, big data and other digital technologies, cross-border information can be obtained to carry out flexible manufacturing, and realize transformation from automated production to intelligent production, from standardized production to personalized production [32]. Some production links are distributed globally through subcontracting or outsourcing, Which helps companies turn to open mode innovation [33]. With rapid development of 5G and other digital technologies, remote service in different places has become a reality. For example, Service providers can provide services in distant foreign countries, there are remote surgery, remote teaching, remote robots in field of manufacturing, etc. Digital technology improves competitiveness by developing digital services to replace traditional services [34].

### 3.2 The resource allocation effect

Digital technology can re-optimize allocation of production factors in traditional manufacturing, it improves efficiency of information transmission, optimize procurement plans, select suitable suppliers, quickly respond to user needs and order distribution networks, and greatly reduce cost of purchasing, production and logistics [35]. Digital technology, which is an input of production factor, has gradually replaced low-end production factor such as low-skilled labor while changing traditional production methods. Under current background of rising labor costs, capital accumulation has been promoted, rate of return on capital and average quality of labor have been improved [36]. For example, the use of industrial robots has improved level of intelligence. Compared with efficiency of capital allocation, the role of digital technology in promoting efficiency of labor allocation is more obvious [22]. Digital technologies such as 5G, Industrial Internet help manufacturing industry to extend to higher value-added services, such as AI with sensors widely collects data through the Internet of Things, timely feedback and information sharing [37]. Manufacturing industry is transforming from product manufacturing to "manufacturing + service", digital technology services replace traditional services, the added value of trade in services and exports increased, and global value chain participation was closer [17].

### 3.3 The penetration effect

As a general technology, the transformative impact of digital technology has quietly penetrated into various fields. In the use of many links such as production, operation, management and

marketing, digital technology has transformed enterprise in all factors, multiple angles and the whole chain, and comprehensively optimized business structure and workflow, its ease of use and diffusion have significantly improved production efficiency and reduced transaction costs [38]. Multinational companies distribute different manufacturing processes in different countries, the more production links and processes, the longer production length of global value chain, and the more difficult it is to cooperate. Application of digital technology to various links facilitates communication and collaboration in value chain links such as products, customers, and markets. Mobile office is more flexible, information communication and project approval are convenient and efficient, and cooperation costs are reduced [39]; The digital technology service platform provides efficient and fast matching for both supply and demand, can find the best partner worldwide. It was found that the use of digital technology broadened information and data resources. Through correlation, new ideas can be discovered, new opportunities can be captured to develop new markets. Multidimensional data can provide support for enterprise decision-making and planning, reducing uncertainty of production [40]. Countries with high digital technology level can efficiently integrated into GVC with cost advantages, and continue to optimize and upgrade, their enterprises are in a more favorable position [21].

In short, continuous breakthrough of new digital technology and its integrated application in real economy have brought profound impact on manufacturing industry chain. With efficient information transmission as the core, transaction and value distribution of manufacturing industry chain have undergone fundamental changes [41]. At the same time, based on characteristics of innovation effect, resource allocation effect and penetration effect of information technology, all links of industrial chain may be upgraded due to empowerment of digital technologies, which will lead to new changes in organization mode, application scenario and consumption demand [18]. Digital technology brings all-round and multiple impacts. Among them, three effects have promoted upgrading of manufacturing industry from different angles. In the three-dimensional driving force, due to influence of new digital technology, transformation of production division of labor, transformation of core competitive advantage and other factors, they will have mutual feedback impact, as shown in Fig 1. Under above overall driving force and interaction mechanism, manufacturing industry upgrading is inevitable [42].

## 4. Model construction, variable selection and data sources

### 4.1 Construction of measurement model

In order to test impact of digital technology on upgrading of the manufacturing industry, the following econometric model is constructed:

$$lnGVC\_PO_{it} = \beta_0 + \beta_1 lnDIG_{it} + \beta_2 Controls + \gamma_i + \mu_t + \delta_{it} \tag{1}$$

The subscript i represents manufacturing industry segment, and t represents the year. The explained variable $GVC\_PO_{it}$ represents international division of labor status of manufacturing industry segments; the core explanatory variable $DIG_{it}$ represents level of manufacturing digitization, and Controls represents control variables, including China's manufacturing global value chain participation (PAT), Research & Development investment level (R&D), Foreign Direct Investment (FDI) and Productivity (PRO), $\gamma_i$, $\mu_t$, and $\delta_{it}$ represent industry fixed effects, year fixed effects and standard error respectively. Taking into account the differences in level values of different variables, in order to reduce influence of heteroskedasticity, the indicators in measurement model are all natural logarithms.

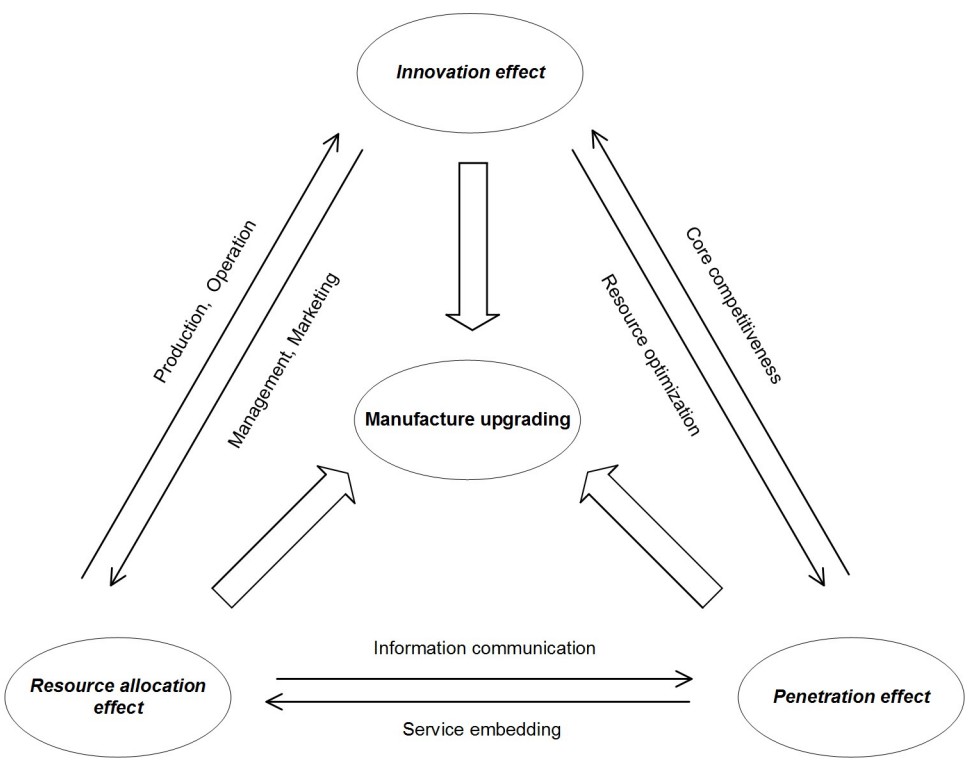

**Fig 1. Mechanism of digital technology affecting manufacturing upgrading.**

## 4.2 Indicator and data sources

**(1) Explained variable.** Global value chain position (GVC_PO), In order to measure status and characteristics of China's manufacturing industry in global value chain, the GVC division of labor status of an industry in a country is measured by referring to global value chain position (GVC_PO) index [43]. The larger the index, the more upstream a country's industry is in global value chain; on the contrary, the lower it is in value chain with a lower division of labor status. The formula is as follows:

$$GVC\_PO = \frac{PLv\_GVC_{ir}}{(PLy\_GVC_{ir})'}$$

Where $PLv\_GVC_{ir}$ represents production length of the country i sector based on forward link. The farther the i sector is from final demand, the greater the value; $PLy\_GVC_{ir}$ represents production length of the country i sector based on backward link, and the i sector is away from all product sectors The farther the distance of initial input end, the greater the value.

**(2) Explaining variables and control variable.** The core explanatory variable is the level of digital technology. Digital technology is the general term for a variety of digital technologies, including hardware, software and network technology. Digital technology is supported by information and communication technologies such as computers, networks and software services, and is supported by automation technologies such as industrial robots and Numerical Control Machine Tools [16]. Through deep integration and full penetration, production efficiency of manufacturing industry can be effectively improved. Therefore, a country's digital technology level can be measured by integration and penetration level of products containing information, communication and automation technologies in manufacturing industry,

Considering connotation and characteristic of digital technology and its impact on upgrading of manufacturing industry, it is defined as computer communication, information and automation and related services [20]. This article draws on idea of using input-output table to calculate direct consumption coefficient to measure the number of digital technology products used per unit of manufacturing output [22]. Select the C26, C28, J61 and J62_J63 sectors as product input value of digital technology industry, and calculate direct consumption coefficient $a_{ij} = x_{ij}/X_j$ for each segment of manufacturing industry ($x_{ij}$ represents the number of products consumed by sector i and $X_j$ represents the total output of sector j). In order to eliminate difference in technology level, the relative value-added rate $v_j = V_j/X_j$ is used to measure the level of technology between industries ($V_j$ is the value-added of the industry, $X_j$ is overall output of the industry). The formula for calculating level of digital technology is:

$$DIG = \sum a_{ij} \times v_j \times 100$$

In order to identify the difference of using domestic digital technology products and foreign digital technology products, index of domestic digital technology products used in China is recorded as DIGD, and index of foreign digital technology products used in China is recorded as DIGF, The total digital technology level index is recorded as DIG. Based on the above calculation method, digital technology level index of China's manufacturing industry from year 2000 to 2014 was calculated as following (Fig 2).

It can be seen from the Fig 2 that, since China joined the WTO in 2001, digital technology has risen rapidly, China has vigorously developed processing trade, and the use of foreign digital technology has increased year by year, although digital technology level index has fluctuated slightly, But the overall trend is increasing. It can be seen from comparison of white and black column that domestic digital technology products used are significantly higher than those of foreign ones. The white column shows domestic DIGD index of China's manufacturing industry increased from 0.267 in 2001 to 0.287 in 2014, with an average annual growth rate of 7.5%.

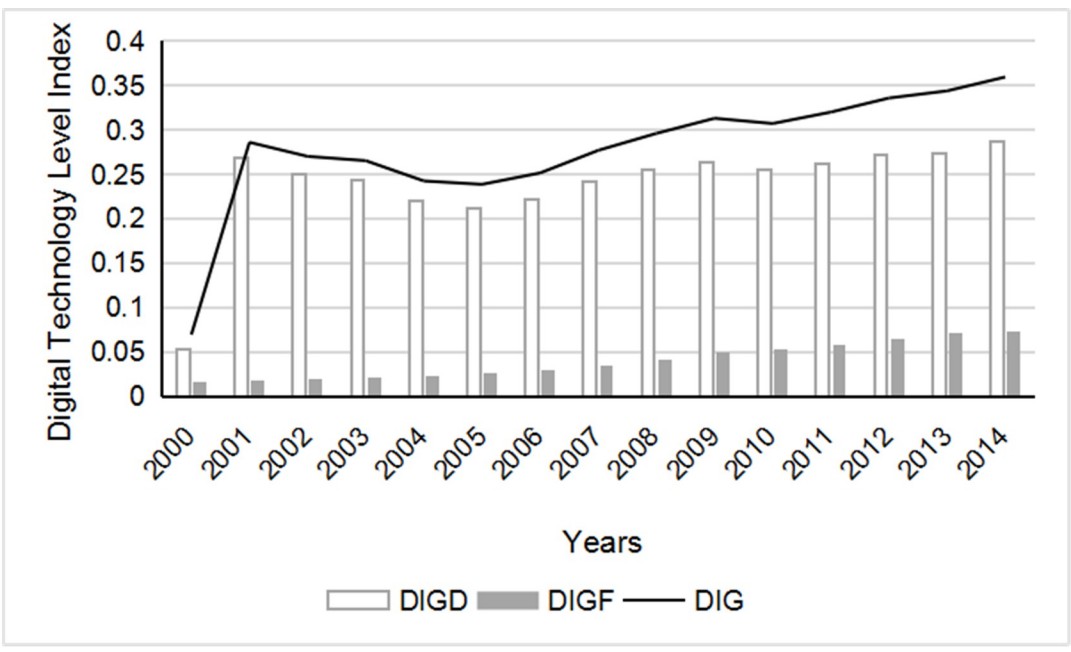

**Fig 2. China's manufacturing digital technology level index.**

The black column displays that foreign DIGF index increased from 0.018 in 2001 to 0.073 in 2014, which was four times that of 2001. The overall DIG index rose from 0.286 in 2001 to 0.359 in 2014, an increase of 0.26 times in 14 years. Despite slight impact of the global financial crisis in 2008, The above figures show that level of digital technology in Chinese manufacturing industry has been continuously improved since 2001, and domestic digital technology products are mainly used. Digital technology has increasingly become an important driving force for development of manufacturing industry [20].

**Control variables** include: Foreign Direct Investment (FDI): measured by proportion of Foreign Direct Investment in GDP; Research & Development input level (R&D): measured by fixed capital of each industry in manufacturing industry divided by the number of employees; productivity (PRO): measured by per capital domestic value added of manufacturing industry; China's manufacturing global value chain participation (PAT) is measured by the sum of GVC forward participation and backward participation.

### 4.3 Data source

China's manufacturing global value chain position, participation, and level of digital technology development are calculated according to WIOD. The total GDP of 43 countries in WIOD accounts for more than 80% of the total world. The latest version of WIOD data is from 2000 to 2014, it contains 56 industry categories, which can better reflect main global economic activities. The database has been widely used due to its continuity and authority [44]. The economic development level and foreign direct investment data come from annual database of China National Bureau of Statistics, and R&D investment and labor productivity are calculated according to social and economic accounts of the WIOD. The descriptive statistical results of each variable are shown in Table 2.

## 5. Empirical test results and analysis

### 5.1 Benchmark model regression

Before regression test, this article uses HT to test stationarity of panel data. The results show that all variables reject assumption of unit roots. The panel fixed effects model (FE) and random effects model (RE) are used to regress Eq (1). The result of Hausman test rejects null hypothesis at the 5% level, so the econometric model adopts a fixed-effects model regression [2]; at the same time, endogenous problem caused by omitted variables is partially solved by controlling the time and industry fixed effects; Columns (1)—(3) of Table 3 report empirical results of core explanatory variable digital technology on manufacturing upgrading, Both China's total digital technology (DIG), domestic digital technology (DIGD) and foreign digital technology (DIGF) used in China have significant positive impact on status of division of

**Table 2. Descriptive statistics of variables.**

| Variable | Obs | Mean | Std. Dev. | Min | Max |
|---|---|---|---|---|---|
| lnGVC_PO | 270 | -0.102 | 0.155 | -0.436 | 0.366 |
| lnDIGD | 270 | 0.239 | 0.166 | 0.007 | 0.665 |
| lnDIGF | 270 | 0.039 | 0.034 | 0.004 | 0.149 |
| lnDIG | 270 | 0.279 | 0.195 | 0.011 | 0.772 |
| lnFDI | 270 | 9.654 | 1.119 | 6.831 | 12.112 |
| lnR&D | 270 | 2.097 | 0.286 | 1.383 | 2.606 |
| lnPRO | 270 | 4.215 | 0.753 | 2.828 | 6.595 |
| lnPAT | 270 | 0.325 | 0.113 | 0.099 | 0.672 |

Table 3. Benchmark regression results.

| Variable | (1) | (2) | (3) | (4) | (5) | (6) |
|---|---|---|---|---|---|---|
| DIGD | 0.131* | | | 0.124* | | |
| | (0.058) | | | (0.052) | | |
| DIGF | | 1.266*** | | | 0.703*** | |
| | | (0.208) | | | (0.201) | |
| DIG | | | 0.160** | | | 0.129** |
| | | | (0.049) | | | (0.045) |
| FDI | | | | -0.067*** | -0.073*** | -0.071*** |
| | | | | (0.0199) | (0.0195) | (0.02) |
| R&D | | | | 0.448*** | 0.390*** | 0.438*** |
| | | | | (0.041) | (0.044) | (0.042) |
| PRO | | | | 0.138*** | 0.135*** | 0.137*** |
| | | | | (0.019) | (0.018) | (0.019) |
| PAT | | | | 0.144 | 0.183 | 0.145 |
| | | | | (0.118) | (0.116) | (0.118) |
| Constant | -0.071*** | -0.085*** | -0.075*** | -0.932*** | -0.766*** | -0.871*** |
| | (0.012) | (0.011) | (0.012) | (0.167) | (0.177) | (0.17) |
| Time fixed effect | Yes | Yes | Yes | Yes | Yes | Yes |
| Sector fixed effect | Yes | Yes | Yes | Yes | Yes | Yes |
| N | 270 | 270 | 270 | 270 | 270 | 270 |
| $R^2$ | 0.115 | 0.219 | 0.135 | 0.467 | 0.481 | 0.473 |

Note: The values in parentheses are standard errors, and ***, **, and * indicate significant at the 1%, 5%, and 10% levels respectively. The following tables are the same.

labor in manufacturing industry, which means that without considering other factors, the higher level of digital technology used in China's manufacturing industry, the more favorable its status in division of labor; other control variables are added in turn, From regression results in (4)—(6) of Table 3, influence coefficient of development level of digital technology on status of manufacturing is still positive significantly, which shows that digital technology can effectively promote upgrading of manufacturing industry, it is consistent with economic expectations. Our results also similar to arguments of He [21] and Liu [22] who believe that digital technology has a positive role.

Although China uses more domestic digital technology, its coefficient of 0.124 is smaller than the coefficient of DIGF (0.703). That is, the use of foreign digital technology can better enhance division of labor status of Chinese manufacturing industry. The possible reason is that foreign digital technology services are more efficient, and developed countries have always occupied a dominant position in automation, internet and other information technology fields, and their digital technology level is in a leading position [45]. China has imported a large number of foreign high-tech intermediate products and used foreign digital technology products, which has a certain technology spillover effect and drives the upgrade. on the other hand, There is little difference in coefficient of DIGF and DIGD, which also shows that China is making rapid progress in field of digital technology, struggling to catch up with developed countries, and the new generation of domestic digital technology is playing an increasingly important role [46]. Gao (2020) analyzed impact differences caused by the use of domestic and foreign digital technology, and suggested that the use of domestic digital technology is more conducive to upgrading of Chinese manufacturing industry [20].

The regression results of main control variables show that: the impact of foreign direct investment (FDI) on manufacturing status is significantly negative at the 1% level. It shows

that low-end locking effect of FDI on manufacturing industry is obvious. Wang (2019) indicated that impact of foreign investment on Chinese manufacturing industry was a "double-edged sword". On the one hand, it promoted integration of Chinese manufacturing into global value chain and build a "world factory", China's manufacturing scale has become the largest in the world, on the other hand, Multinational companies rely on brand and technology control, firmly grasp dominance of global value chain, implement technology blockade and reduce profit margins of Chinese products for their own interests, and "lock" China's manufacturing industry in low-end position of global value chain [47, 48]. R&D investment is conducive to improvement of innovation and technological level, and promotes transformation of China's manufacturing industry from original equipment manufacturer (OEM) to high-value-added links such as R&D and design. Strengthening R&D investment is an important factor for many developing countries to achieve value chain improvement at certain stage [49, 50]. Digital technology can improve labor productivity, The increase of labor productivity is conducive to enhance manufacturing added value, regression results prove its significant positive effect. Since participation includes forward participation and backward participation, that is, importing intermediate products for processing or exporting intermediate products to other countries, China mainly imports intermediate products for processing & assembly and then export to other countries, such as importing parts and components to produce Apple mobile phones, China gets only a tiny profit, which is not conducive to promotion of status [51]. Although GVC participation of China's manufacturing industry is high, it is facing dilemma of low-end locking, regression coefficient of value chain participation has not passed significance test, Liu (2020) also proved this argument, It is necessary to improve domestic value added in exports and climb to the middle and high end of global value chain [25].

## 5.2 Robustness test

In order to verify robustness of the above results, the following three methods are used to test reliability of empirical results. **First**, the method of replacing core variable index is adopted, and the upstream degree index constructed by Antràs (2018) is used as a substitute index of GVC_PO to regress [52]. The results are reported in columns (1)—(3) of Table 4. The coefficients of DIGD, DIGF and DIG were 0.125, 0.223 and 0.105 respectively, which were all significant at 1% level; The effects of other control variables are the same as before. The results show

**Table 4. Robustness test results.**

| Variable | (1) FE | (2) FE | (3) FE | (4) 2SLS | (5) 2SLS | (6) 2SLS |
|---|---|---|---|---|---|---|
| DIGD | 0.125*** (0.034) | | | 0.543*** (0.055) | | |
| DIGF | | 0.223** (0.138) | | | 2.180*** (0.386) | |
| DIG | | | 0.105*** (0.030) | | | 0.478*** (0.048) |
| FDI | -0.038** (0.013) | -0.029* (0.013) | -0.039** (0.013) | -0.139*** (0.008) | -0.137*** (0.011) | -0.145*** (0.008) |
| R&D | 0.173*** (0.027) | 0.153*** (0.030) | 0.165*** (0.027) | 0.373*** (0.028) | 0.343*** (0.030) | 0.371*** (0.027) |
| PAT | 0.160* (0.078) | 0.193* (0.079) | 0.166* (0.078) | 0.063 (0.056) | 0.168* (0.067) | 0.087 (0.055) |
| Constant | 1.940*** (0.111) | 1.894*** (0.122) | 1.963*** (0.114) | -0.137 (0.072) | -0.001 (0.115) | -0.069 (0.076) |
| Time fixed effect | Yes | Yes | Yes | Yes | Yes | Yes |
| Sector fixed effect | Yes | Yes | Yes | Yes | Yes | Yes |
| Kleibergen-Paap rk LM statistic | | | | 63.514 [0.0000] | 53.969 [0.0000] | 54.602 [0.0000] |
| Cragg-Donald Wald F statistic | | | | 481.465 | 4657.614 | 235.884 |
| Kleibergen-Paap rk Wald F statistic | | | | 336.145 | 3928.454 | 114.523 |
| N | 270 | 270 | 270 | 252 | 252 | 252 |
| $R^2$ | 0.940 | 0.937 | 0.940 | 0.610 | 0.558 | 0.612 |

that core explanatory variable digital technology is still significantly positive, indicating that the above results are robust. **Second**, using instrumental variable method [26], participation of manufacturing industry in global value chain may also have an impact on its digitization level. In order to prevent the two-way causality between digital technology and status of manufacturing division of labor, overcome potential endogeneity, and reduce lagging level of digitization level. The lag period of digital level is introduced into equation as its instrumental variable, which can eliminate influence of current period to a certain extent. The underidentification test strongly rejects null hypothesis of "the instrumental variable is unrecognizable", and weak instrumental variable test rejects null hypothesis at the 1% level, indicating that instrumental variable is reasonable; correlation test result of instrumental variable and endogenous variable shows that F statistics are greater than 10 in the first stage, and adjusted $R^2$ is significant at the 1% level. The test shows that 2SLS regression can be used. The regression results are reported in columns (4)—(6) of Table 4. It can be seen that estimated coefficients of variables (0.543,2.180,0.478) have all passed the 1% significance level test, and the estimated signs are consistent with benchmark regression results, further indicating robustness of the above test results.

**Third**, quantile regression. The quantile regression estimation is more robust, and it can more comprehensively analyze overall characteristics of digital technology affecting division of labor in manufacturing industry. The regression results of digital technology level at quantile points of 0.1, 0.5, 0.9 are given below, More accurately describe the change trend of digital technology level and manufacturing status at different quantiles. Table 5 shows that impact of digital technology at each quantile is significantly positive, and as quantile increases from 0.1,0.5 to 0.9, coefficients of DIGD (0.578,0.401,0.210), DIGF (2.027,2.280,0.009), DIG (0.481,0.386,0.220) gradually decrease respectively, which indicates that impact of digital technology on distribution of manufacturing upgrade conditions is gradually reduced. The possible reason is that, compared with promoting upgrading of the manufacturing industry in the higher end of the value chain, digital technology has a greater effect on manufacturing industry at the lower end of the value chain, China's manufacturing industry, which is at the lower end of the global value chain, can effectively utilize digital technology and improve its position through digital transformation [25].

**Table 5. Quantile regression results.**

| Variable | (1) | (2) | (3) | (4) | (5) | (6) | (7) | (8) | (9) |
|---|---|---|---|---|---|---|---|---|---|
| | QR_10 | | | QR_50 | | | QR_90 | | |
| DIGD | 0.578*** (0.050) | | | 0.401*** (0.087) | | | 0.210** (0.066) | | |
| DIGF | | 2.027*** (0.341) | | | 2.280*** (0.564) | | | 0.009* (0.414) | |
| DIG | | | 0.481*** (0.047) | | | 0.386*** (0.077) | | | 0.220*** (0.061) |
| FDI | -0.094*** (0.008) | -0.126*** (0.011) | -0.101*** (0.009) | -0.121*** (0.015) | -0.134*** (0.018) | -0.130*** (0.015) | -0.099*** (0.011) | -0.071*** (0.013) | -0.108*** (0.012) |
| R&D | 0.276*** (0.027) | 0.331*** (0.027) | 0.283*** (0.028) | 0.295*** (0.047) | 0.270*** (0.045) | 0.289*** (0.046) | 0.325*** (0.036) | 0.201*** (0.033) | 0.333*** (0.036) |
| PRO | 0.042*** (0.008) | 0.055*** (0.009) | 0.039*** (0.008) | 0.091*** (0.01) | 0.074*** (0.015) | 0.085*** (0.014) | 0.141*** (0.011) | 0.157*** (0.011) | 0.136*** (0.011) |
| PAT | 0.153* (0.063) | 0.373*** (0.066) | 0.172** (0.065) | 0.049 (0.109) | 0.184 (0.111) | 0.084 (0.107) | -0.208* (0.083) | -0.207* (0.081) | -0.187* (0.084) |
| Constant | -0.266*** (0.070) | -0.155 (0.096) | -0.206** (0.076) | -0.034 (0.121) | 0.178 (0.160) | 0.057 (0.124) | -0.281** (0.092) | -0.306** (0.117) | -0.209* (0.098) |
| N | 270 | 270 | 270 | 270 | 270 | 270 | 270 | 270 | 270 |

## 5.3 Further analysis

**(1) Factor density heterogeneity analysis.** The upgrade differences of manufacturing sectors are closely related to characteristics of factor intensity. In order to analyze differences in manufacturing upgrades with different factor intensities, manufacturing industries in the WIOD are divided into three categories. Sectors C5—C9, C16, and C22 are labor-intensive manufacturing industries, sectors C10—C15 are capital-intensive manufacturing industries, and sectors C17—C21 are technology-intensive manufacturing industries [53]. We test impact of digital technology on manufacturing upgrades in groups. The regression results are shown as follows in Table 6.

As can be seen from Table 6 that digital technology has sector difference in upgrading of manufacturing industries with different factor intensiveness. In labor-intensive industries, impact of using domestic digital technology products on manufacturing upgrades is significant at the 10% level, but impact of using foreign digital technology products is not significant, and the total effect of DIG is not significant either. Compared with labor-intensive industries, digital technology can promote status of capital-intensive and technology-intensive industries better, The promotion of foreign digital technology is more obvious. The possible reason is that investment of digital technology in labor-intensive industries is relatively low, If digital technology such as industrial robot replace labor, it will have significant impact on labor-intensive industries [54]. Therefore, the above table shows that effect of digital technology on labor-intensive industries is not significant. Digital technology requires more capital. AI, industrial robots and other digital technologies are used differently in different factor intensive sectors, and are more commonly used in capital intensive and technology intensive industries. The flexibility of digital technology to replace traditional elements is higher in capital-intensive and technology-intensive industries, and its effect on production and operation efficiency is more obvious [47].

China's traditional manufacturing industry is the most competitive sector, mainly labor-intensive industries, The application of digital technology may bring great impacts and challenges, weakening China's factor cost advantage, Developed countries use industrial robots and other technologies to transform manufacturing links, resulting in backflow of manufacturing industry, which will inhibit upgrading of China's manufacturing industry [26]. Huang Qunhui(2019) put forward views that under background of "the third industrial revolution",

**Table 6. Factor density heterogeneity analysis.**

| Variable | (1) | (2) | (3) | (4) | (5) | (6) | (7) | (8) | (9) |
|---|---|---|---|---|---|---|---|---|---|
| | Labor-intensive | | | Capital-intensive | | | Technology-intensive | | |
| DIGD | 0.123* (0.110) | | | 0.0818** (0.090) | | | 0.133** (0.074) | | |
| DIGF | | 0.936 (0.511) | | | 1.060*** (0.280) | | | 0.140** (0.429) | |
| DIG | | | 0.134 (0.098) | | | 0.124** (0.075) | | | 0.108** (0.066) |
| Constant | -1.572*** (0.329) | -1.184** (0.411) | -1.493*** (0.342) | -0.972*** (0.272) | -0.972*** (0.236) | -0.922*** (0.266) | 0.093 (0.404) | -0.036 (0.445) | 0.105 (0.411) |
| Time Fixed Effect | Yes | Yes | Yes | Yes | Yes | Yes | Yes | Yes | Yes |
| Sector Fixed Effect | Yes | Yes | Yes | Yes | Yes | Yes | Yes | Yes | Yes |
| N | 105 | 105 | 105 | 90 | 90 | 90 | 75 | 75 | 75 |
| $R^2$ | 0.426 | 0.441 | 0.430 | 0.670 | 0.726 | 0.679 | 0.758 | 0.743 | 0.755 |

replacement of simple labor force by automation and intelligence of production may have an impact on China's comparative labor cost advantage, and may also block China's transformation and upgrading path from comparative advantage to competitive advantage [55]. It is necessary to seize the opportunities provided by digital technology, transform and upgrade traditional industries, seize commanding height of digital transformation, and accelerate integration and innovation of digital technology and manufacturing industry in a wider and deeper extent. The digital technology level of technology-intensive industries is relatively high. Digital transformation needs to overcome technical bottleneck, accelerate expansion and application of new technologies such as Industrial Internet and AI in field of intelligent manufacturing, transform technical advantages into industrial advantages, and provide new driving force for high-quality development of manufacturing industry [56].

(2) **Technology level heterogeneity analysis.** Manufacturing sectors with different technical levels are also affected by digital technology. According to Li Fuyu's (2018) technology classification method, the C5—C9 and C22 sectors in WIOD database are classified as low-tech manufacturing industries, and the C10, C13—C15 sectors are classified as low-to-medium technology manufacturing industry, sectors C11, C12, C17—C21 are classified as medium-to-high technology manufacturing industries [57]. Group regression tests impact of digital technology on upgrading of manufacturing industries with different technical levels. The results are shown in Table 7 below.

The results in Table 7 show that there are differences in impact of digital technology on manufacturing with different technological levels. Low-tech industries have not passed significance test, and digital technologies are not as significant as expected. Perhaps because low-tech industries are mostly resource-intensive industries, too much digital input may offset labor cost advantage and weaken positive effect [8]; As far as low-to-medium technology industry and medium-to-high technology industry are concerned, input of digital technology has a significant positive effect, They are significant at the 1% or 5% level respectively. For example, China's high-speed rail has widely used digital technologies such as Beidou navigation, mobile communications, artificial intelligence and the Internet of Things to optimize transportation, management, operation, and entire decision-making process, Digital technology has promoted the high-end rise of high-speed rail value chain [58]; among them, impact of foreign digital technology products is particularly significant. It may be that imports of foreign intermediate products are critical to development of Chinese manufacturing industry, for example China is highly dependent on imports of foreign chips; There are also medium-to-high tech industries

**Table 7. Technology level heterogeneity analysis.**

| Variable | (1) | (2) | (3) | (4) | (5) | (6) | (7) | (8) | (9) |
|---|---|---|---|---|---|---|---|---|---|
| | Low technology | | | Low-to-medium technology | | | Medium-to-high technology | | |
| DIGD | 0.076 (0.118) | | | 0.127** | | | 0.157** | | |
| | | | | (0.129) | | | (0.063) | | |
| DIGF | | 0.719 (0.567) | | | 1.470*** (0.389) | | | 0.547** (0.362) | |
| DIG | | | 0.088 (0.107) | | | 0.177** (0.105) | | | 0.138*** (0.056) |
| Constant | -1.844*** (0.364) | -1.522** (0.463) | -1.784*** | -1.040** | -1.186*** | -0.989** | -0.289 | -0.235 (0.346) | -0.243 (0.300) |
| | | | (0.381) | (0.375) | (0.312) | (0.360) | (0.292) | | |
| Time Fixed Effect | Yes | Yes | Yes | Yes | Yes | Yes | Yes | Yes | Yes |
| Sector Fixed Effect | Yes | Yes | Yes | Yes | Yes | Yes | Yes | Yes | Yes |
| N | 90 | 90 | 90 | 75 | 75 | 75 | 105 | 105 | 105 |
| $R^2$ | 0.452 | 0.462 | 0.454 | 0.472 | 0.580 | 0.490 | 0.744 | 0.732 | 0.744 |

greatly affected by digitization. For example, digital technology is applied to aircraft simulation design, The production process, measurement, inspection and quality management data are presented on three-dimensional model, which speeds up development progress and greatly reduces research and development costs [12]. Shi Bo (2020) believes that the use of digital technologies such as artificial intelligence can promote innovation, bring technology spillover, cultivate high-end production factors and form high-end capital [59]. The input of digital production factors is conducive to integrating into global value chain and expanding space for GVC participation. The application of digital technology is crucial to improvement of its quality and efficiency, and climbing effect of global value chain is becoming more and more obvious [60, 61].

## 6. Conclusions and recommendations

Digital technologies have profound impact on manufacturing global value chain. With economic model of China's manufacturing industry shifting from high-speed growth to high-quality development, empowerment through digital technology is an inevitable choice for future transformation and upgrading. This paper uses 2000–2014 data of WIOD to measure development level of China's digital technology, It is found that digital technology had developed rapidly since China's entry into WTO in 2001. Theoretical and empirical analysis shows:

(1) The effect of digital technology on upgrading of Chinese manufacturing industry is significantly positive; Distinguish digital technology sources found that the use of domestic digital technology accounts for a larger proportion, while foreign digital technology services are more efficient, and both have promoted upgrading of manufacturing industry; The empirical test shows that FDI has dual effects of pulling effect and low-end locking effect. Strengthening R&D plays an important role in breaking through technology blockade; (2) Distinguish impact of industries with heterogeneous factor intensity and technological level, it is found that there were industry differences. Digital technology plays more significant role in promoting industries with high technical level and high capital intensity. The above conclusions have important implications for development of Chinese digital technology.

The first is to accelerate process of digital transformation of manufacturing industry. China needs to fully rely on domestic digital elements to cultivate a flexible and lean production system and break through "low-end locking" trap. It is necessary to consolidate digital technology development foundation of new infrastructure, and constantly upgrade infrastructure such as 5G, big data, cloud computing and artificial intelligence, We should strengthen R&D and innovation of key common technologies such as high-performance chips and basic software, and get rid of dependence of core technologies on foreign countries as soon as possible. Manufacturing enterprises need to jointly build a collaborative innovation system, establish digital industry clusters with international competitiveness, and form a new open pattern.

The second is scientific dealing with important impact of foreign digital elements on the control of domestic enterprises. How to rely on digital trade to fully release spillover effect of foreign digital elements is a practical issue worth considering. China advocates liberalization of digital trade, actively participates in negotiation of digital trade rules, and builds a rule system in line with interests of developing countries. Taking the "Belt and Road" initiative as an opportunity, build partnership with countries along the route, promote interconnection and optimal allocation of digital resources such as big data centers and 5G base stations, and deepen digital technology collaboration.

The third is to implement differentiated policies according to the industry. Labor-intensive industries make full use of digital transformation promoted by domestic digital technology, and deepen application of digital technology in production, operation, management, and

after-sales services, to achieve quality transformation and efficiency improvement. Advanced manufacturing industry should strengthen R&D and innovation, use emerging information technologies such as AI and the Internet of Things to transform business operations and product service models, and take development path of digitization and intelligence.

## Supporting information

**S1 Data.**
(XLSX)

## Author Contributions

**Writing – original draft:** Qingwei Fu.

**Writing – review & editing:** Qingwei Fu.

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
