## [Decision Letter · Decision Letter 0]

17 Jan 2022

PONE-D-21-40013How Does Digital Technology Affect Manufacturing Upgrading?Theory and Evidence from ChinaPLOS ONE

Dear Dr. Fu,

Thank you for submitting your manuscript to PLOS ONE. After careful consideration, we feel that it has merit but does not fully meet PLOS ONE’s publication criteria as it currently stands. Therefore, we invite you to submit a revised version of the manuscript that addresses the points raised during the review process.

We look forward to receiving your revised manuscript.

Kind regards,

Mohammad Azadi

Academic Editor

PLOS ONE

Journal Requirements:

2. We note you have included a table to which you do not refer in the text of your manuscript. Please ensure that you refer to Table 4 in your text; if accepted, production will need this reference to link the reader to the Table.

3. Please include a copy of Table 9 which you refer to in your text.

Additional Editor Comments:

1) The first part of the article is written without any references.

2) The novelty of this manuscript should be highlighted compared to the literature review.

3) All formulations need references, unless they were extracted by authors.

4) What is the reason for reporting data in Figure 1 until 2014? What are data for recent years?

5) Data in Table 1 is unclear. Numbers are not in a good format. Check it again please. Such a problem could be seen for other tables.

6) R2 values are so low in Table 2. What is the reason? They could not be used for a correct analysis. Such a problem could be seen in some other cases in other tables. Best fitting should be bolded.

7) The number of references should be extended for a better discussion. Obtained results should be compared to other results of other researches.

Reviewers' comments:

Reviewer's Responses to Questions

**Comments to the Author**

1. Is the manuscript technically sound, and do the data support the conclusions?

Reviewer #1: Partly

Reviewer #2: Partly

2. Has the statistical analysis been performed appropriately and rigorously? 

Reviewer #1: Yes

Reviewer #2: Yes

3. Have the authors made all data underlying the findings in their manuscript fully available?

Reviewer #1: Yes

Reviewer #2: No

4. Is the manuscript presented in an intelligible fashion and written in standard English?

Reviewer #1: Yes

Reviewer #2: No

5. Review Comments to the Author

Reviewer #1: I appreciate the author's efforts and the novelty of his ideas on the submitted manuscript, but there are still some issues that the author should consider in order to further improve the manuscript.

—There were some errors in sentence structure and punctuation, as well as problems with table formatting. The language of the manuscript still needs improvement. In order to truly contribute to a valuable article, the author should explore related field research more deeply, and put forward the profound thoughts of contribution and innovation of this paper compared with the current research status in the first and second parts of the article. Therefore, I think the author needs to do more work, especially to revise and improve the manuscript.

—The last part of the introduction should be a summary of the manuscript research, which is a clear summary of what will be studied. It mainly restates the key steps and main ideas of the paper and shows how they are proved. I believe this will greatly increase the degree of future citation of this literature in related fields.

—Although the author has made some theoretical explorations in the third section, in my opinion, the analysis in this part is not in-depth enough. The author should consider the deeper mechanism of digital technology to promote the upgrading of manufacturing industry, and make this part more theoretical support. In addition, I believe that if the author can change the research hypothesis and put forward the statement, it will make readers understand the relevant research content more clearly.

—The manuscript has made abundant robustness verification for the regression of the benchmark model and has also carried out heterogeneity analysis from different angles, which is worthy of affirmation. However, some theoretical analysis of the results is not thorough enough, especially in the case of insignificant regression results. In my opinion, detailed, reasonable and in-depth exploration can add depth to the article and explore the development of related fields.

—The conclusions and suggestions are short, and the suggestions are not targeted enough .I think if this point can be improved, the research will be more oriented.

Reviewer #2: Relevance. The paper investigates the theoretical mechanism of digital technology for manufacturing upgrading. This paper incorporates digital technology and the global value chain division of labor into a unified analytical framework. On the one hand, digital technology's contribution is evaluated from domestic and foreign perspectives; on the other hand, the heterogeneity of manufacturing factor intensity and the classification of technology levels are used to deepen the understanding and elaboration of the relationship between digital technology for manufacturing upgrading. The research in this paper is enlightening and guiding researchers to deepen their understanding of the impact of the digital economy.

Rigour. The econometric model in this paper is reasonably and scientifically constructed and can adequately support the research topic of this paper. However, the following improvements in the research discussion need to be considered.

(1) Some expressions are inappropriate and even confusing. For example, there is a problem with the expression "1% high" in the third line of the text on page 15 of the article. Please explain the meaning of 1% in detail.

(2) Please further concise and highlight the semantic precision in the presentation of the contribution of the article.

(3) Although the data sources are explained in this paper, digitalization in China is developing rapidly. Therefore, the use of data in this paper seems to be old and the research data is rather lagging, please consider adding the latest data after 2014.

(4) Figure 1 shows the digital technology level index of China's manufacturing industry from 2000 to 2014, but the analysis of the interpretation of Figure 1 is not sufficient, and some rich descriptions expressed by the columns in the figure should be given.

(5) The text display of Table 1and Table 2 is further optimized. it currently feels like there is a problem with the typography.

Impact: This work has the potential to have a notable impact as it presents the theoretical mechanism of digital technology for manufacturing upgrading and gives the results and description of the empirical analysis.

Quality of presentation and writing. The paper followed a structure, and the quality of written communication was generally good. However, some improvements still need to be made. The authors need to optimize further the presentation of the article in terms of the various sections such as the abstract, introduction, model, empirical evidence, discussion, and especially the illustration of the content of the figures and tables. Very importantly, the authors could consider native English speakers to improve the quality of the language presentation of this article. Currently, the potential value of the article is obscured due to language presentation issues.

6. PLOS authors have the option to publish the peer review history of their article (what does this mean?). If published, this will include your full peer review and any attached files.

Reviewer #1: No

Reviewer #2: No

---

## [Author Response · Author response to Decision Letter 0]

22 Feb 2022

Response to Reviewers

I sincerely thank the editor and all reviewers for the valuable feedback that I have used to improve the quality of my manuscript.I have made extensive corrections to the previous paper according to your nice suggestions. I`ll be happy to edit paper further,based on helpful comments from the reviewers.The detailed corrections are listed below.

1.Please ensure that your manuscript meets PLOS ONE's style requirements, including those for file naming. 

Response:I have revised article according to the style requirements and submitted the 3 files.

2.We note you have included a table to which you do not refer in the text of your manuscript. Please ensure that you refer to Table 4 in your text; if accepted, production will need this reference to link the reader to the Table.

Response:I tried to explain Quantile regression results,now revised in the paper.

3. Please include a copy of Table 9 which you refer to in your text.

Response:I have corrected “Table 9” to “Table 5”

Additional Editor Comments:

1) The first part of the article is written without any references.

Response:The first part add 7 references now. 

2) The novelty of this manuscript should be highlighted compared to the literature review.

Response:Compared with the current research,the novelty was highlighted in the paper. 

3) All formulations need references, unless they were extracted by authors.

Response:References have been added and marked.

4) What is the reason for reporting data in Figure 1 until 2014? What are data for recent years?

Response:The paper uses the data of WIOD2016 which covering the year from 2000 to 2014.Currently the data is from major transnational input-output databases include WIOD, TiVA, GTAP-ICIO and ADB-MRIO, which cover different countries, industrial sectors and time spans (see following table).

Database Issuing agency Covered

Country or region Covered

sector Time 

WIOD2013 EU 40 35 1995-2011

WIOD2016 EU 43 56 2000-2014

WTO-OECD TiVA WTO 61 34 1995,2000,2005,2008-2011

GTAP-ICIO  Purdue University 121 43 2004、2007、2011

ADB-MRIO  ADB 63 35 2000，2007-2019

The world input-output database (WIOD) has been updated to 2014. The latest version of WIOD (2016) contains continuous time series data from 2000 to 2014, covering 43 countries / regions and 56 industry categories, and includes statistical data of environmental and socio-economic accounts. The total GDP of countries and regions accounts for more than 80% of the world, which can objectively reflect major global economic activities, It is widely used by the international research scholars because of its continuity and authority. 

I added data of 2016-2019 in the first section in order to illustrate the current situation of China's digital economy.

5)Data in Table 1 is unclear. Numbers are not in a good format. Check it again please. Such a problem could be seen for other tables.

 Response:The format of Table 1 and Table 2 have been revised.

6) R2 values are so low in Table 2. What is the reason? They could not be used for a correct analysis. Such a problem could be seen in some other cases in other tables. Best fitting should be bolded.

 Response:The main objective of empirical test using panel data is to determine which explanatory variables are significant, test whether the impact of DIG on GVC-PO is significantly correlated, and the value of R-squared will not affect the interpretation of the relationship between DIG and GVC-PO. The R2 value is relatively small when the explanatory variable is not added as the control variable, After the explanatory variable is added, the R2 value becomes larger, indicating that the model interpretation is acceptable.

Similar situations have been encountered in the literatures such as follows:the R2 is 0.01 or 0.167981

Source：Antras P.,Chor D.,Fally T.,Hillberry R.Measuring the Upstreamness of Production and Trade Flows[J].American Economic Review,2012,102(3),412～416.

Source：Wang PeiZhi, Muhammad Ramzan.Do corporate governance structure and capital structure matter for the performance of the firms? An empirical testing with the contemplation of outliers[J].Plos one, February 27, 2020 https://doi.org/10.1371/journal.pone.0229157

7) The number of references should be extended for a better discussion. Obtained results should be compared to other results of other researches.

 Response:The number of references have been extended from 24 to 39,Obtained results had been compared to other results of other researches in the paper now.

Reviewer #1:

—There were some errors in sentence structure and punctuation, as well as problems with table formatting. The language of the manuscript still needs improvement. In order to truly contribute to a valuable article, the author should explore related field research more deeply, and put forward the profound thoughts of contribution and innovation of this paper compared with the current research status in the first and second parts of the article. Therefore, I think the author needs to do more work, especially to revise and improve the manuscript.

 Response:Thank you for your careful review. I have improved the language and compared with the current research in the first and second parts of the article

—The last part of the introduction should be a summary of the manuscript research, which is a clear summary of what will be studied. It mainly restates the key steps and main ideas of the paper and shows how they are proved. I believe this will greatly increase the degree of future citation of this literature in related fields.

 Response:The last part of the introduction restated the key steps and main ideas of the paper and showed how they are proved.

—Although the author has made some theoretical explorations in the third section, in my opinion, the analysis in this part is not in-depth enough. The author should consider the deeper mechanism of digital technology to promote the upgrading of manufacturing industry, and make this part more theoretical support. In addition, I believe that if the author can change the research hypothesis and put forward the statement, it will make readers understand the relevant research content more clearly.

 Response:In the third section,I tried deeper mechanism and changed the research hypothesis and put forward the statement.

—The manuscript has made abundant robustness verification for the regression of the benchmark model and has also carried out heterogeneity analysis from different angles, which is worthy of affirmation. However, some theoretical analysis of the results is not thorough enough, especially in the case of insignificant regression results. In my opinion, detailed, reasonable and in-depth exploration can add depth to the article and explore the development of related fields.

 Response:I tried thorough analysis,especially in the case of insignificant regression results in paper now.

—The conclusions and suggestions are short, and the suggestions are not targeted enough .I think if this point can be improved, the research will be more oriented.

Response:I revised conclusions and suggestions in the paper.

Reviewer #2

(1) Some expressions are inappropriate and even confusing. For example, there is a problem with the expression "1% high" in the third line of the text on page 15 of the article. Please explain the meaning of 1% in detail.

Response:Thank you for the suggestions.***, **, and * indicate significant at the 1%, 5%, and 10% levels respectively. All the tables are the same.Now I revised as follows:”the impact of Foreign Direct Investment(FDI) on the status of the division of labor in the manufacturing industry is negative, which is significant at the 1% level” on page 15 mentioned.

(2) Please further concise and highlight the semantic precision in the presentation of the contribution of the article.

 Response:I tried to highlight the contribution of the article in the paper now.

(3) Although the data sources are explained in this paper, digitalization in China is developing rapidly. Therefore, the use of data in this paper seems to be old and the research data is rather lagging, please consider adding the latest data after 2014.

 Response:I added the latest data in the first section.

(4) Figure 1 shows the digital technology level index of China's manufacturing industry from 2000 to 2014, but the analysis of the interpretation of Figure 1 is not sufficient, and some rich descriptions expressed by the columns in the figure should be given.

 Response:I further explained and illustrated Figure 1 and descriptions expressed by columns in the paper now.

(5) The text display of Table 1and Table 2 is further optimized. it currently feels like there is a problem with the typography.

 Response:Table 1 and Table 2 is optimized.

---

## [Decision Letter · Decision Letter 1]

9 Mar 2022

PONE-D-21-40013R1How Does Digital Technology Affect Manufacturing Upgrading?Theory and Evidence from ChinaPLOS ONE

Dear Dr. Fu,

Thank you for submitting your manuscript to PLOS ONE. After careful consideration, we feel that it has merit but does not fully meet PLOS ONE’s publication criteria as it currently stands. Therefore, we invite you to submit a revised version of the manuscript that addresses the points raised during the review process.

We look forward to receiving your revised manuscript.

Kind regards,

Mohammad Azadi

Academic Editor

PLOS ONE

Journal Requirements:

Additional Editor Comments (if provided):

1) Formulations have no references. Before the formulation, at the end part of the last sentence (before the formulation), references should be mentioned.

2) It is better to add a quantitative results in the abstract.

3) The title of the third part in the manuscript should be shortened. It is too lengthy.

Reviewers' comments:

Reviewer's Responses to Questions

**Comments to the Author**

1. If the authors have adequately addressed your comments raised in a previous round of review and you feel that this manuscript is now acceptable for publication, you may indicate that here to bypass the “Comments to the Author” section, enter your conflict of interest statement in the “Confidential to Editor” section, and submit your "Accept" recommendation.

Reviewer #1: (No Response)

Reviewer #2: All comments have been addressed

2. Is the manuscript technically sound, and do the data support the conclusions?

Reviewer #1: (No Response)

Reviewer #2: Yes

3. Has the statistical analysis been performed appropriately and rigorously? 

Reviewer #1: (No Response)

Reviewer #2: Yes

4. Have the authors made all data underlying the findings in their manuscript fully available?

Reviewer #1: (No Response)

Reviewer #2: Yes

5. Is the manuscript presented in an intelligible fashion and written in standard English?

Reviewer #1: (No Response)

Reviewer #2: Yes

6. Review Comments to the Author

Reviewer #1: I appreciate the author's efforts and the novelty of his ideas on the submitted manuscript, but there are still some issues that the author should consider in order to further improve the manuscript.

1.There are some errors in sentence structure and punctuation in parts of the manuscript, and parts of the manuscript are repeated. Authors should carefully examine the grammatical structure and make further modifications to the language of the manuscript.

2.Literature Review in Part II,the author discusses the impact of digital technology on manufacturing upgrades from the definition, theoretical and empirical perspectives of digital technology. This approach is worthy of recognition. However, the authors define digital techniques in the manuscript. In the following micro-level research, I think the author should be highly integrated with the definition given by himself.

3.Although the author has done some theoretical exploration in the third part, the analysis in this part is not deep enough in my opinion. The author proposes three effect mechanisms of digital technology on the manufacturing value chain, but how are these three effects proposed, and whether there is some connection between the three. In this regard, the authors should conduct further exploration.

4.The manuscript has made abundant robustness verification for the regression of the benchmark model and has also carried out heterogeneity analysis from different angles, which is worthy of affirmation. However, the author's previous statement mentioned the current situation of unbalanced development of China's manufacturing regions, whether it is possible to consider the differences between regions in the heterogeneity analysis.

Reviewer #2: The authors have understood the reviewers' comments more accurately and have answered and revised several important questions more accurately.

7. PLOS authors have the option to publish the peer review history of their article (what does this mean?). If published, this will include your full peer review and any attached files.

Reviewer #1: No

Reviewer #2: No

---

## [Author Response · Author response to Decision Letter 1]

19 Mar 2022

Response to Reviewers

I sincerely thank the editor and reviewers for the valuable feedback that I have used to improve the quality of my manuscript. I have made corrections to the previous paper according to your nice suggestions. Enclosed please find the responses to the reviewers. I sincerely hope this manuscript will be finally acceptable to be published. Thank you very much for all your help and looking forward to hearing from you soon.

1.Formulations have no references. Before the formulation, at the end part of the last sentence (before the formulation), references should be mentioned. 

Response: Thanks for the suggestion. References have been mentioned and added from 39 to 51 and marked.

2.It is better to add a quantitative result in the abstract

Response: Thanks for the suggestion. I`ve added quantitative results in the abstract.

3.The title of the third part in the manuscript should be shortened. It is too lengthy.

Response: Thanks for the suggestion. It was shortened.

Reviewer #1:

1.There are some errors in sentence structure and punctuation in parts of the manuscript, and parts of the manuscript are repeated. Authors should carefully examine the grammatical structure and make further modifications to the language of the manuscript. 

Response: Thank you for your suggestion. The spelling,punctuation errors, repeated parts have been checked and corrected.

2.Literature Review in Part II,the author discusses the impact of digital technology on manufacturing upgrades from the definition, theoretical and empirical perspectives of digital technology. This approach is worthy of recognition. However, the authors define digital techniques in the manuscript. In the following micro-level research, I think the author should be highly integrated with the definition given by himself.

 Response: In the micro-level research, I have integrated with the definition. I defined digital technology in the fourth part now.

3.Although the author has done some theoretical exploration in the third part, the analysis in this part is not deep enough in my opinion. The author proposes three effect mechanisms of digital technology on the manufacturing value chain, but how are these three effects proposed, and whether there is some connection between the three. In this regard, the authors should conduct further exploration.

Response:I made further exploration with the figure 1 to explain three effect mechanisms of digital technology on the manufacturing value chain.

4.The manuscript has made abundant robustness verification for the regression of the benchmark model and has also carried out heterogeneity analysis from different angles, which is worthy of affirmation. However, the author's previous statement mentioned the current situation of unbalanced development of China's manufacturing regions, whether it is possible to consider the differences between regions in the heterogeneity analysis.

Response: Thanks for your good suggestion. I gave further explaining about the differences between regions. Due to the limitation of data, the heterogeneity analysis of regional differences cannot be carried out. I hope to use new data to do further research in the future paper.

---

## [Decision Letter · Decision Letter 2]

28 Mar 2022

PONE-D-21-40013R2How Does Digital Technology Affect Manufacturing Upgrading?Theory and Evidence from ChinaPLOS ONE

Dear Dr. Fu,

Thank you for submitting your manuscript to PLOS ONE. After careful consideration, we feel that it has merit but does not fully meet PLOS ONE’s publication criteria as it currently stands. Therefore, we invite you to submit a revised version of the manuscript that addresses the points raised during the review process.

We look forward to receiving your revised manuscript.

Kind regards,

Mohammad Azadi

Academic Editor

PLOS ONE

Journal Requirements:

Additional Editor Comments (if provided):

Please carefully address the reviewers' comments besides the attachments and also the following comments,

1) The abbreviation WIOD should be defined at first mentioning.

2) No quantitative results could be found in the abstract.

3) The title of parts is lengthy. They should be shortened.

4) All formulations need references, unless they were extracted by authors.

5) The conclusion part is lengthy. It should be also shortened.

Reviewers' comments:

Reviewer's Responses to Questions

**Comments to the Author**

1. If the authors have adequately addressed your comments raised in a previous round of review and you feel that this manuscript is now acceptable for publication, you may indicate that here to bypass the “Comments to the Author” section, enter your conflict of interest statement in the “Confidential to Editor” section, and submit your "Accept" recommendation.

Reviewer #1: (No Response)

2. Is the manuscript technically sound, and do the data support the conclusions?

Reviewer #1: (No Response)

3. Has the statistical analysis been performed appropriately and rigorously? 

Reviewer #1: (No Response)

4. Have the authors made all data underlying the findings in their manuscript fully available?

Reviewer #1: (No Response)

5. Is the manuscript presented in an intelligible fashion and written in standard English?

Reviewer #1: (No Response)

6. Review Comments to the Author

Reviewer #1: I am grateful to the author for adequately revising the comments I made in the last round.I appreciate the author's efforts and the novelty of his ideas on the submitted manuscript, but there are still some issues that the author should consider in order to further improve the manuscript.

1.In terms of language expression, I think the author should pay attention to his own way of expression. In the manuscript, some expressions are inappropriate, and the semantic precision should be highlighted.

2.In the second part of the literature review, the author has sorted out a very good framework, which is very worthy of recognition from the perspectives of definition, theory and evidence.However, the author should sort out more cutting-edge literature in related fields (such as citing literature in the past three years many times). In this manuscript, some of the literature is relatively old.

3.In the fourth part, the author lists China's manufacturing digital technology level index, whether it is possible to consider adding updated data.

7. PLOS authors have the option to publish the peer review history of their article (what does this mean?). If published, this will include your full peer review and any attached files.

Reviewer #1: No

---

## [Author Response · Author response to Decision Letter 2]

4 Apr 2022

Response to Reviewers

I sincerely thank editor and reviewer for the valuable feedback that I have used to improve quality of my manuscript. I have made corrections to previous paper once more. Enclosed please find the responses to reviewers. I sincerely hope manuscript will be acceptable soon. 

1) The abbreviation WIOD should be defined at first mentioning. 

Response: Thanks for the suggestion. World Input-Output Database abbreviation WIOD was defined at first mentioning now.

2) No quantitative results could be found in the abstract.

Response: Thanks for the suggestion. I added quantitative results in the abstract.

3) The title of parts is lengthy. They should be shortened.

Response: Thanks for the suggestion. Title was shortened.

4)All formulations need references, unless they were extracted by authors.

Response: Thanks for the suggestion. References have been mentioned and added.

5)The conclusion part is lengthy. It should be also shortened.

Response: Thanks for the suggestion. Conclusion part was shortened.

Reviewer #1:

1.In terms of language expression, I think the author should pay attention to his own way of expression. In the manuscript, some expressions are inappropriate, and the semantic precision should be highlighted. 

Response: Thank you for your suggestion. The expressions were checked and corrected, the semantic precision was highlighted.

2.In the second part of the literature review, the author has sorted out a very good framework, which is very worthy of recognition from the perspectives of definition, theory and evidence.However, the author should sort out more cutting-edge literature in related fields (such as citing literature in the past three years many times). In this manuscript, some of the literature is relatively old.

 Response: Thank you for your suggestion.The older literatures are deleted and the latest references are added.

3.In the fourth part, the author lists China's manufacturing digital technology level index, whether it is possible to consider adding updated data.

Response: Thanks for your good suggestion. Since the digital technology level index of China's manufacturing industry is calculated according to the World Input-Output Database, and data in the database is from year 2000 to 2014. As a remedy,updated data about digital economy was added in the first part. It is hoped to do further research in the future.

---

## [Editor Report · Decision Letter 3]

6 Apr 2022

How Does Digital Technology Affect Manufacturing Upgrading?Theory and Evidence from China

PONE-D-21-40013R3

Dear Dr. Fu,

We’re pleased to inform you that your manuscript has been judged scientifically suitable for publication and will be formally accepted for publication once it meets all outstanding technical requirements.

Kind regards,

Mohammad Azadi

Academic Editor

PLOS ONE

Additional Editor Comments (optional):

It is accepted. However, please remove the references in the conclusions part. They should be moved to the discussion part in the proof stage.
---

## [Editor Report · Acceptance letter]

22 Apr 2022

PONE-D-21-40013R3 

How does digital technology affect manufacturing upgrading? Theory and evidence from China 

Dear Dr. Fu:

I'm pleased to inform you that your manuscript has been deemed suitable for publication in PLOS ONE. Congratulations! Your manuscript is now with our production department. 

Kind regards, 

on behalf of

Dr. Mohammad Azadi 

Academic Editor

PLOS ONE